# Shallow Permafrost at the *Crystal* Site of Peaceful Underground Nuclear Explosion (Yakutia, Russia): Evidence from Electrical Resistivity Tomography

Svetlana Artamonova [1,*], Alexander Shein [2,3,*] , Vladimir Potapov [3], Nikolay Kozhevnikov [3] and Vladislav Ushnitsky [4]

[1] V.S. Sobolev Institute of Geology and Mineralogy, Russian Academy of Sciences, Siberian Branch, 630090 Novosibirsk, Russia

[2] Science Center for Arctic Research, Yamal-Nenets Autonomous District, 629008 Salekhard, Russia

[3] A.A. Trofimuk Institute of Petroleum Geology and Geophysics, Russian Academy of Sciences, Siberian Branch, 630090 Novosibirsk, Russia; PotapovVV@ipgg.sbras.ru (V.P.); kozhevnikovno@ipgg.sbras.ru (N.K.)

[4] Ministry of Ecology, Nature Management and Forestry of the Sakha Republic (Yakutia), 677000 Yakutsk, Russia; ushnitski@mail.ru

[*] Correspondence: artam@igm.nsc.ru (S.A.); A.N.Shein@yandex.ru (A.S.)

**Abstract:** The site where a peaceful underground nuclear explosion, *Crystal*, was detonated in 1974, at a depth of 98 m in perennially frozen Cambrian limestones, was studied by electrical resistivity tomography (ERT) in 2019. The purpose of our research, the results of which are presented in this article, was to assess the current permafrost state at the *Crystal* site and its surroundings by inversion and interpretation of electrical resistivity tomography (ERT) data. Inversion of the ERT data in *Res2Dinv* verified against *ZondRes2D* forward models yielded 2D inverted resistivity sections to a depth of 80 m. The ERT images revealed locally degrading permafrost at the *Crystal* site and its surroundings. The warming effect was caused by two main factors: (i) a damage zone of deformed rocks permeable to heat and fluids, with a radius of 160 m around the emplacement hole; (ii) the removal of natural land cover at the site in 2006. The artificial cover of rock from a nearby quarry, which was put up above the emplacement hole in order to prevent erosion and migration of radionuclides, is currently unfrozen.

**Keywords:** environment research; peaceful underground nuclear explosion; electrical resistivity tomography; permafrost; geological environment; natural–technical system; Yakutia

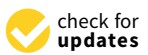



## 1. Introduction

The impact of underground nuclear explosions (UNE) on the Earth's crust is the most violent man has ever caused [1]. The extreme heat and pressure of an underground nuclear explosion causes changes in the surrounding rocks: those closest to the source are molten and vaporized, forming a confinement cavity with a hardpan of melt rocks. Farther away, zones of crushed and cracked rocks are created [1]. Compression waves reflected from the flat surface of the lithosphere–atmosphere boundary produce a zone of spalling (at hundreds of ms), shaped as a sphere segment around epicenter. Shallow explosions cause swelling and failure of rocks as upthrust domes that then fall back due to gravity. Then, the cavity collapses, and the crushed rocks rise ("float up"), forming a rubble chimney in seconds to hours [1]. The collapsed confinement cavity surrounded by heavily deformed rocks becomes an uncontrolled storage source of special radioactive wastes in the geological environment.

Since 1945, many nuclear explosions have been conducted at training test sites in deserted areas. Underground nuclear explosions were also used for peaceful economic purposes, following special government programs: 27 explosions by the USA from 1961

to 1973 and 124 by the former USSR between 1965 and 1988 [2]. Most of the peaceful underground nuclear explosions (PUNE) in the USSR were detonated outside the training test sites (Figure 1), often in populated residential or mining areas. In this respect, studies of PUNE sites' subsurfaces are indispensable to mitigate the risks from the leakage and migration of radionuclides.

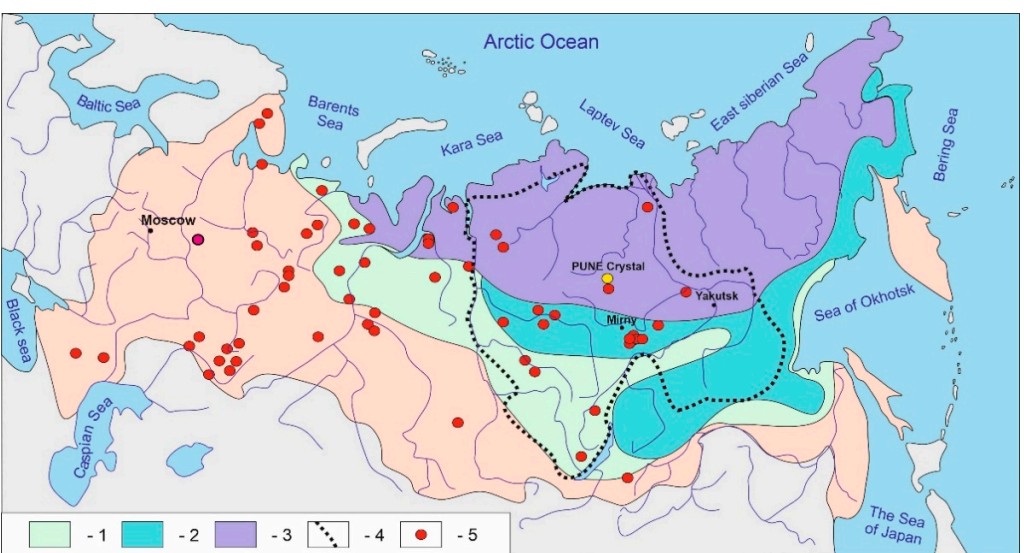

**Figure 1.** Location of *Crystal* site (yellow circle). Blue shades show continuous (1), discontinuous (2), and sporadic (3) permafrost. Dotted line (4) contours the Siberian craton. Red circles (5) mark PUNE sites in the former Russian Federation.

About a half of the 623 publications found by keyword search for underground nuclear explosion at http://apps.webofknowledge.com (last accessed on 29 August 2021) [3] deal with explosion sites, while the others focus more on the detection of nuclear tests by seismic and radiochemical methods [3]. The hazardous PUNE sites actually arouse great interest in geoscientists but the data for such research may be of confidential restricted access, and thus, the number of publications is not significant.

In Russia, PUNE sites became open to research after disclosure in the early 1990s. Most of the studies dealt with radioactivity patterns in surface landscapes [4–6]; few works discussed the PUNE impact on rocks inferred from groundwater [7,8], seismic [9], and resistivity [10–12] surveys. The latter included transient electromagnetic (TEM) soundings at two PUNE sites, *Crystal* and Craton-3, located in permafrost [10–12].

Most of the peaceful underground nuclear explosions in Russia (51 out of 85) were conducted in taiga and tundra landscapes of northeastern Russia (Figure 1), in permafrost.

Permafrost is a natural barrier to fluid circulation in shallow subsurface. According to some scientists [2], permafrost would obstruct the propagation of radionuclides from the explosion source to the ground surface. Meanwhile, permafrost is extremely vulnerable to external impacts. The naturally existing and new PUNE-induced zones of deformed rocks that are permeable to heat and fluids can interfere with the original heat budget, cause permafrost degradation, and allow radionuclides to leak through the thaw windows. Underground nuclear explosions produce mixed natural–technical systems specific to each site with their own histories of response to the strong temperature, pressure and radiation impacts depending on local geology and technical blasting conditions.

The subsurface of the PUNE sites in Russia remains underexplored, but the knowledge of their settings is crucial as a basis for hazard prediction and mitigation, especially in populated permafrost areas. The resistivity section of permafrost is high contrast and can be studied by resistivity surveys, while non-destructive remote sensing methods are preferable at the PUNE sites. The purpose of our research, the results of which are presented in this

article, was to assess the current permafrost state at the *Crystal* site and its surroundings by inversion and interpretation of electrical resistivity tomography (ERT) data.

## 2. Site *Crystal* of Peaceful Underground Nuclear Explosion

The *Crystal* PUNE site is located near the mined Udachny kimberlite pipe (Yakutian diamond province, Siberian craton, Russia) and mine-related Udachny community in Yakutia: 4 km and 8 km away, respectively (Figure 1).

Sediments in the area lie upon an Archean basement and are intruded with kimberlites. The sedimentary sequence (bottom to top) comprises interbedded thin Ediacaran carbonate and clastic rocks (~200 m thick); Cambrian limestone, dolomite, and their silty and clayey varieties, totaling a thickness of 2200 m; a ≤200 m thick discontinuous layer of Ordovician marine facies; and Quaternary soft sediments mostly restricted to depressed landforms, with thicknesses of only 2 m on slopes and watersheds and 10 m in valleys. The emplacement hole's stripped shallow subsurface is composed of Quaternary clay silt and debris (0–5 m); heavily fractured marl (5–9.8 m); only slightly deformed clay-free limestone (9.8–21.5 m); and interbedded deformed clayey and dolomitic limestones (21.5–105.3 m) [11].

The study area is a hilly terrain of sparce taiga landscape, with Larix gmelinii growing on slopes [13]. Continuous ice-rich permafrost reaches a thickness of 150 to 200 m and lies over the Upper Cambrian and first Middle Cambrian aquifers that enclose lenses of brines at negative temperatures (cryopegs). In the 1980s, the 0 °C isotherm was as deep as 800 to 1050 m [14], and the total permafrost thickness was about 1 km, mainly due to very low deep heat flux (from 13 to 27 mW/m$^2$), as determined by V. Balobaev in the 1970s [14]. The deeper subpermafrost aquifers are second Middle Cambrian, Lower Cambrian and Neoproterozoic. Both Middle Cambrian aquifers are especially rich in brines with salinity up to 450 g/L. The confining pressure increases with depth and is 1.0–1.5 to 26 MPa in excess of the normal hydrostatic value [8,15].

According to the original project, eight shallow PUNE along a linear profile traversing the Ulakhan-Bysyttakh River valley were supposed to upthrust rock domes and produce mounds rising 27–30 m high above the surface (PUNEs of this type were described as "loosening" explosions [2]). The mounds were meant to make an 1800 m long and 85 m wide (on the crest) dam and a small pond that would retain the tailings from the Udachny diamond mine. The first explosion (*Crystal*), with a yield of 1.7 kt of TNT equivalent, was detonated on 2 October 1974 on the left side of the river, 98 m below the ground, in perennially frozen limestone.

The explosion produced an upthrust dome with a diameter of 160–180 m, which rose to a maximal height of 60 m (Figure 2a) in 3.5 s after detonation (Figure 2a) and then settled back due to gravity. Strong underground thumps were noted 17 h after the explosion. Finally, the dome-shaped mound was formed with a diameter of 200 m and an average height of 10 m above the original surface (locally 14 m in the western side), which was twice as low as anticipated (Figure 2b). For that official reason, the remaining seven explosions were cancelled.

Parameters of rock deformation at the *Crystal* site (Figure 2) were estimated with reference to patterns of UNE-induced rock failure studied in field experiments [1]. The radius of the confinement cavity is related to the explosion yield (E, kt) as $R_{cav} \sim a_1^{-1} E^{1/3}$ (the factor $a_1$ is proportional to the rock strength and the buried depth of the device). The radius $R_{cav}$ of the *Crystal* PUNE was about 9.5–14.6 m [1,11,16] (Figure 2b). The damage zone consisted of a crushed zone, with its radius ($R_{crush}$) directly proportional to $E^{1/3} a_2 R_{cav}$ (factor $a_2$ depends on the strength and compressibility of rocks and is 4.2–5.3 for the geological conditions of the *Crystal* site), $R_{crush} = E^{1/3} \cdot (4.2–5.3) R_{cav} \sim 48–92$ m; a cracked zone, with the radius ~105 m ($R_{crack} = 2R_{crush}$) exceeding the burial depth and extending to the ground surface; and a spalling zone of the radius $R_{spall}$ on the land around the epicenter, estimated ~160 m from the depth and yield of the explosion as $R_{spall} = W + H_{spall}$, where W is the burial depth, $H_{spall}$ is the spalling depth, $H_{spall} = C_p \cdot \tau_+ / 2$, C and $\tau_+$ are, respectively, the velocity and arrival time of compression waves [1]. The upthrust dome rose at a rate

of 34 m/s for 3.5 s [1], while the spalling depth was 59.5 m and the spalling radius was ~160 m (Figure 2b).

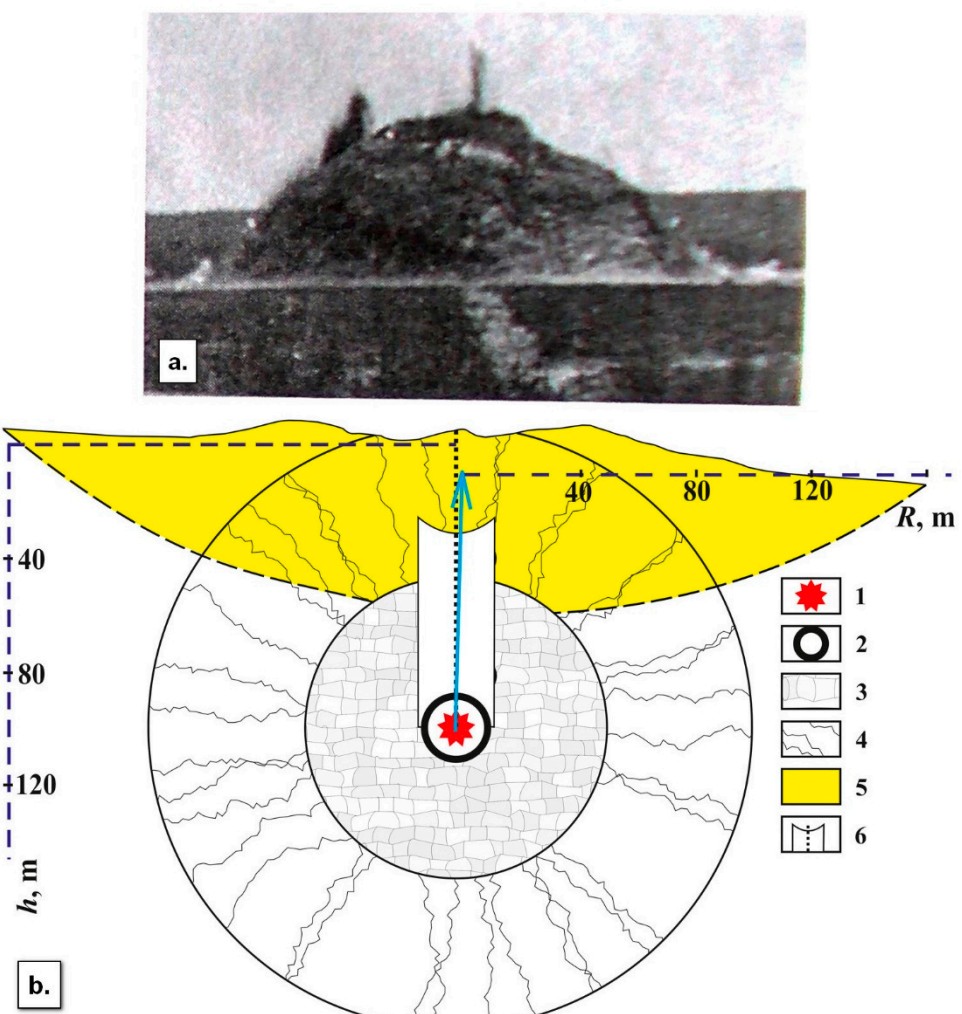

**Figure 2.** (**a**): Snapshot of the PUNE *Crystal* site about 3.5 s after detonation, when the upthrust dome reached its maximum height of ~60 m, at a base width of 200 m (the PUNE *Crystal* filming was conducted from a location situated south–south-west of the explosion epicenter [1]). (**b**): Sketch of the confinement cavity with crushed and cracked zones around. (1) Shot point; (2) confinement cavity with a hardpan of molten rock; (3) crushed zone; (4) cracked zone; (5) spalled zone, with contours of the dome-shaped mound, according to the field topographic surveys; (6) chimney (dashed line in the middle is the emplacement hole). Arrow shows the direction of shock pressure wave.

The rubble chimney resulting from the cavity collapse seventeen hours after the explosion was 72–98 m high: the height related to the cavity radius and the explosion yield was $((6–8) \cdot R_{cav} \cdot E^{1/3})$ [1].

Since the blasting works toward the ground surface, all zones of deformed rocks became superposed one upon another to form the one integral damage zone around the emplacement hole (Figure 2). At a small slope of the surface, the explosion epicenter almost coincided with the emplacement hole head (turquoise arrow in Figure 2b).

In 1992, the original the dome-shaped mound was filled with limestone and dolomite debris (dead rocks from the Udachny kimberlite quarry) and topped up further in the winter of 2006, to make up an artificial cover above the explosion hole head and the dome-shaped mound. The artificial cover, 260 m in diameter and 20 m high (Figure 3), was installed with the aim of preventing erosion of the dome-shaped mound and migration of radionuclides [1,4]. In December 2006, the land cover (all plants and soil to 10–15 cm

depth) was removed from the *Crystal* site over an area of 330 m × 430 m, except for the southwestern part, and the site was fenced with barbwire on 4 m high metal stakes planted 2 m deep under the ground (Figure 3).

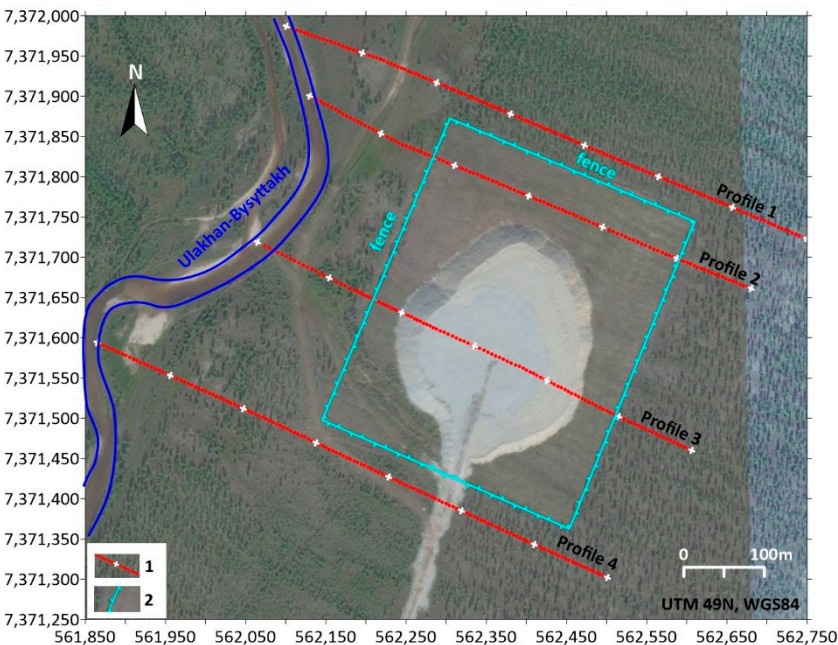

**Figure 3.** ERT profiles (1) of 2019 at and around the *Crystal* site on a Google map; (2) the fence of the site. The artificial cover above the dome-shaped mound and embankment road are in light gray.

Field surveys of 2008–2009, 2012 and 2018–2019 by teams from the V.S. Sobolev Institute of Geology and Mineralogy SB RAS and the A.A. Trofimuk Institute of Petroleum Geology and Geophysics SB RAS (Novosibirsk) revealed leakage of brackish water bearing chlorides and radionuclides from beneath the artificial cover [11,17–19]. The explosion epicenter was marked by a local low resistivity anomaly detected by TEM soundings, which was attributed to the rise of saline fluids through the damage zone around the emplacement hole [10,11].

## 3. ERT Data Acquisition and Inversion

Electrical resistivity tomography (ERT) monitoring is widely used in permafrost studies [20–22]. At the *Crystal* PUNE site, ERT surveys were first carried out in 2019 along four profiles (Figure 3), using the SKALA-48 multielectrode system (designed jointly by teams from the Electrometria design office and the A.A. Trofimuk Institute of Petroleum Geology and Geophysics SB RAS, Novosibirsk, Russia). The system, with its multicore cables (streamers), switches successively to different combinations of 5 m-spaced electrodes (grounded metal rods), which are connected to 48 streamer outputs (Figure 4). The resistivity structure along the profiles was imaged by an offset pole–dipole array, with one current electrode set 1 km off the profile, and a dipole–dipole array, which was used in profile 3 for checking the data quality and resolution (Figure 4). The two arrays ensured penetration to depths of 80 m and 40 m, respectively. In total, 2620 m of profiles were collected, plus 300 m of repeated measurements.

The data quality was checked by measuring the relative rms error (data with >1% error were culled out) and by replica measurements, with an average reproducibility no worse than 5%. Then, the raw resistivity data were processed manually. The ERT data inversion was carried out using the *Res2Dinv* program, with terrain correction, using the conventional Gauss–Newton robust method. As recommended in the *Res2Dinv* manual [20], the inversion was carried out on a dense grid with cells, the size of which was half the distance between the electrodes. This grid spacing improved the resolution of shallow subsurface responses affected by the metal fence and buried iron fragments. The 2D

automatic inversion yielded resistivity of layers and 2D images of the subsurface along the profiles to a depth of 80 m. The images were verified against *ZondRes2D* forward models [23], and the obtained synthetic data were inverted in *Res2DInv* [20].

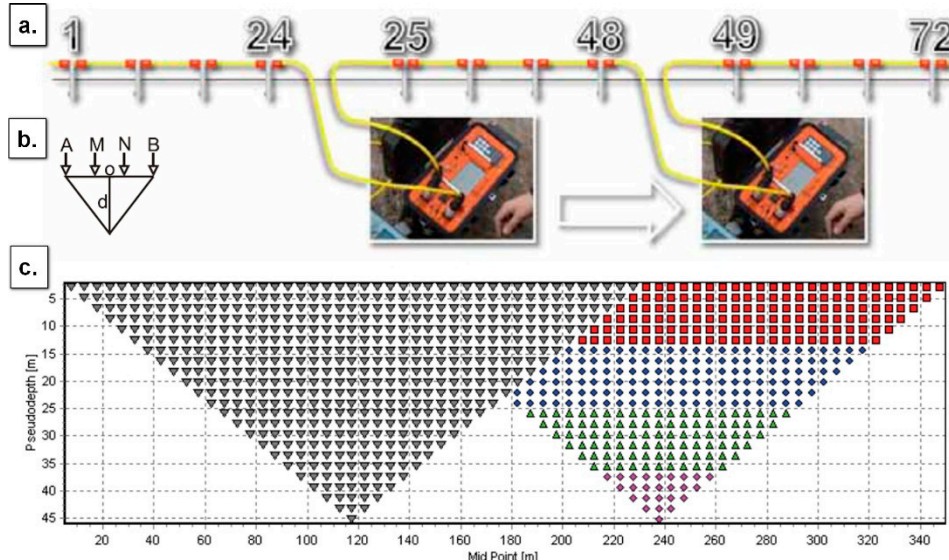

**Figure 4.** ERT measurements. (**a**): Electrodes and cables connected to the SKALA system. (**b,c**): Symmetrical sounding array (**b**) and resistivity data points (**c**).

## 4. Results

The permafrost of the area was studied along four ERT profiles that ran in the SEE–NWW direction (Figure 3), from a forested slope of the Ulakhan-Bysyttakh left bank, at an elevation about 330 m asl, then downslope into the river valley, to 300 m asl. Profiles 1 and 4 were 715 m long each, and were located outside the PUNE site, among Larix gmelinii taiga: 40 m north of the fence and 300 m far from the emplacement hole (profile 1); 30 m south of the fence and 200 m from the emplacement hole (profile 4).

Profiles 2 and 3 were 595 m long each, traversed the *Crystal* site, and crossed twice the fence. As with profile 4, profile 2 was located 200 m away from the emplacement hole and 30 m from the artificial cover, in the northern part of the site where grass and small shrubs had grown spontaneously in the place of the removed land cover. Profile 3 ran in the middle of the artificial cover put up above the emplacement hole.

### 4.1. Forested Slope and River Valley (Profiles 1 and 4)

The ERT surveys along profiles 1 and 4 made it possible to assess the state of the permafrost under conditions of nearly undisturbed (profile 1) and slightly disturbed (profile 4) natural landscape; profile 1 was found to cross only the winter road, whereas profile 4 crossed the winter road and the embankment road (Figure 3). The inverted resistivity section of the slope segment between 0 m and 430 m along profile 1 consisted of three layers (Figure 5a): the 1–2 m thick layer I of low resistivity (145 to 1000 Ohm·m, or locally up to 2000 Ohm·m); the 15–30 m layer II of high resistivity (7400 to 27,000 Ohm·m, locally up to 100,000 Ohm·m or more); and layer III with relatively high resistivity of 2000–7400 Ohm·m, locally 280 to 1000 Ohm·m, up to the maximum penetration depth (80 m).

The river valley segment between 430 and 715 m along profile 1 showed high resistivity up to the 80 m depth, from 7400 Ohm·m or higher beneath the thin conductive layer I. The high-resistivity zone encloses conductive patches of 2000 to 75 Ohm·m in the 470–510 m segment, the lower section part (Figure 5a), and within 145 Ohm·m near the river (segment 690–715 m), from 0 to 7 m depths.

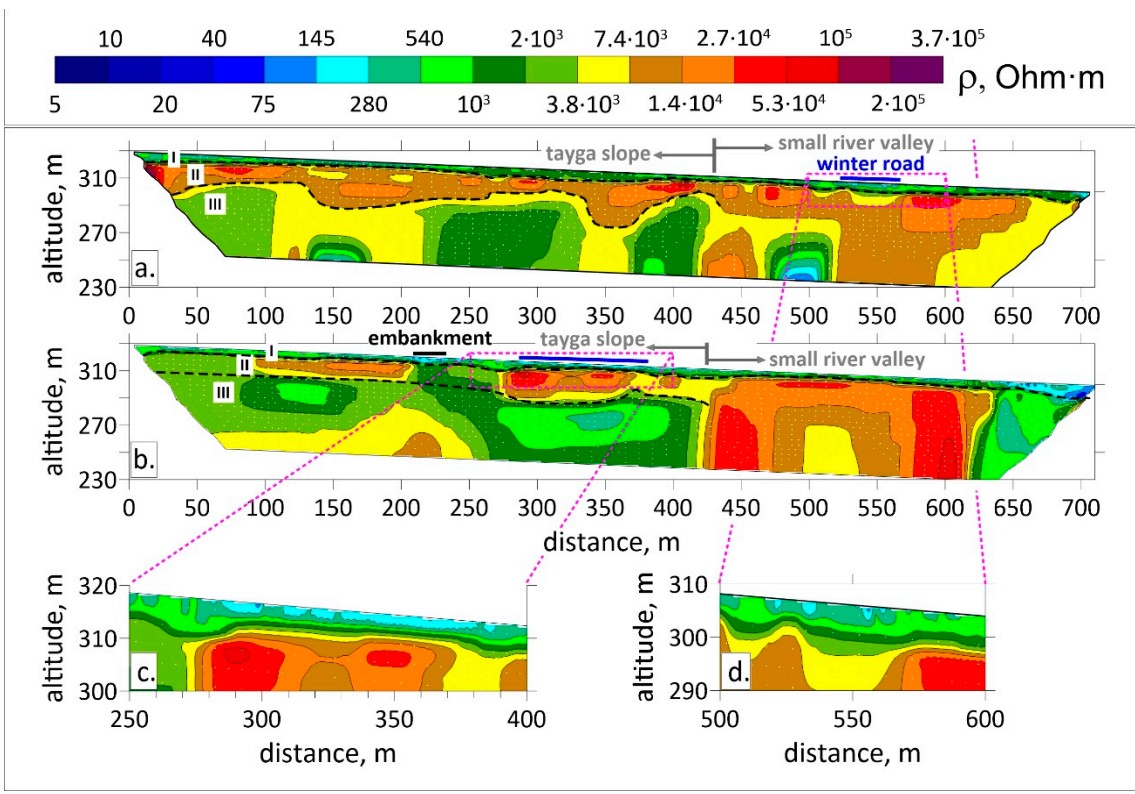

**Figure 5.** Inverted resistivity section (Ohm·m) along profile 1 (**a**) and 4 (**b**), and their large-scale insets within winter road cross segments (**c,d**). Complete pole–dipole measurements. White dots are resistivity data points used in inversion.

The inverted resistivity section of profile 4 was found to be similar to that of profile 1 (Figure 5b), with three resistivity layers beneath the slope (segment 0–410 m), but layer II was less prominent while layer III enclosed low-resistivity patches of 280–540 to 1000 Ohm·m, as well as a nearly vertical 1000–2000 Ohm·m zone merging with layer III, beneath the embankment road.

The inverted resistivity section in the river valley segment of profile 4 consisted of the thin conductive layer I above the high-resistivity subsurface, mostly 7400 to 100,000 Ohm·m (Figure 5b), except for slightly lower values (3800 Ohm·m) within 500 to 540 m, below 40 m. The resistivity was low near the river, in the 620 to 715 m segment: from 1000 Ohm·m in the lower part of the section to 40 Ohm·m in the shallow part. The transition between the high- and low-resistivity zones was sharp and almost vertical.

### 4.2. Northern Part of the Crystal Site (Profile 2)

The inverted resistivity section along the profile 2 was quite intricate (Figure 6a), but shared some features of similarity with that of profile 1. It included the low-resistivity top layer I (280–540 Ohm·m), which was 2–3 m to 4–5 m thick; the 20–30 m thick layer II (1000 to 3800 Ohm·m, or locally to 14,000 Ohm·m) in the slope segment, at 0 to 390 m; and a high-resistivity zone of 7400 to 27,000 Ohm·m or higher reaching the section base, within the 390–595 m segment (river valley). However, unlike the pattern of the undisturbed landscape, the high-resistivity zone enclosed a large conductor with an extremely low resistivity, decreasing from 145 down to 5 Ohm·m with depth.

The extreme low of 10–40 Ohm·m produced by the fence, as well as the anomalies below it, widened with depth, while the resistivity increased. At the depth of 30–40 m, the anomalies merged with 40 to 280 Ohm·m conductive patches.

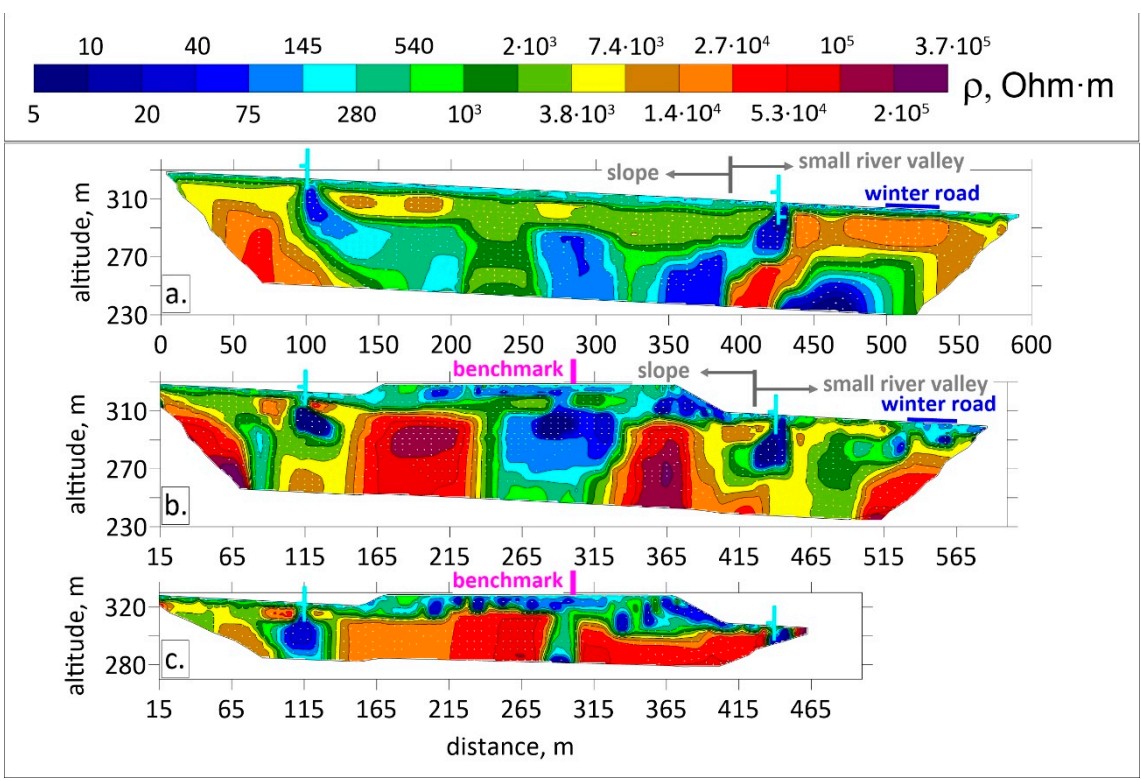

**Figure 6.** Inverted resistivity section (Ohm·m) along profiles 2 (**a**) and 3 (**b,c**). Pole–dipole (**a,b**) and dipole–dipole (**c**) arrays.

### 4.3. Artificial Cover (Profile 3)

Profile 3 crosses the artificial cover above the emplacement hole and, thus, represents the most heavily disturbed part of the *Crystal* site (Figure 6b). The inverted resistivity section within the 0–100 m profile segment corresponding to the natural landscape outside the fence was similar to those along profiles 1 and 4, with a 2–3 m thick layer I of 280–1000 Ohm·m, whereas the resistivity inside the fence within the site was heterogeneous.

The upper ~20 m of the inverted resistivity section (corresponding to the artificial cover thickness) was low resistive: 20 to 280 Ohm·m, though being up to 540 Ohm·m and higher locally. In the middle of the profile, the resistivity was extremely low (5–75 Ohm·m) at depths from 20 to 60 m, but increased to 280–540 Ohm·m at larger depths. The conductor was about 80 m wide and was surrounded with highly resistive rocks of 27,000–53,000 or even up to 200,000 Ohm·m.

The 420–595 m profile segment within the river valley showed low to moderate resistivity of 540 to 3800 Ohm·m up to the section base, with two small coalescent resistivity lows near the river: 40–540 Ohm·m at a depth of 0–22 m in segment 515–535 and 40–280 Ohm·m at 0–13 m between 535 m and 575 m. Local anomalies of extremely low resistivity also appeared at 30–40 m depth beneath the fence, such as in profile 2.

In order to improve the lateral resolution and to check the data quality, the measurements along profile 3 were additionally performed with a dipole–dipole array, which was sensitive to vertical interfaces [24,25] but could penetrate to depths that were twice as shallow. The pole–dipole and dipole–dipole arrays originated at the same point, but the control profile was shorter (475 m). The dipole–dipole inverted resistivity section (Figure 6c) was generally similar to that of the main profile (Figure 6b), with the same sizes and contours of the anomalies beneath the fence, though the central anomaly was only 30 m wide, instead of 80 m, possibly because of the shallower penetration depth.

## 5. Discussion

### 5.1. Permafrost along Profiles 1 and 4: Slopes and Ulakhan-Bysyttakh Valley

The ERT surveys were performed at positive air temperatures in August–September 2019. Therefore, the low-resistivity layer I (140–1000 or locally up to 2000 Ohm·m), 1–2 m or locally 3 m thick, was interpreted as the active layer, and is known to vary in thickness from 0.3 to 1.7 m on the local slopes [13]. The 1–2 m thickness of ERT layer I agrees with the values based on geocryological and geothermal data. Thus, the resistivity under 2000 Ohm·m can be attributed to unfrozen rocks.

The slope and river valley inverted resistivity sections differed: layered permafrost was found in the former and high resistivity was identified below the active layer down to 80 m in the latter, with low-resistivity patches near the river. The active layer was commonly thinner while the permafrost was colder in the bottom of small river valleys and on northern slopes than on other slopes and watersheds: 0.2–0.9 m as compared to 0.3–1.7 m [8,14], and −4.5 to −8.0 °C as compared to −2 to −3.5 °C [8], respectively. Thus, the initial permafrost conditions in the valleys of small rivers, such as the Ulakhan-Bysyttakh, were more prominent than on the slopes. This inference is consistent with temperature measurements of 2001 in a borehole located 300 m west of the emplacement hole, which proved the preservation of −4.9 to −5.4 °C permafrost at depths from 5 m to 180 m, despite the proximity to the PUNE site [8].

The high-resistivity features of 7400 Ohm·m and higher, detected beneath the active layer in the river valley 350 m NNW and 260 m SSW of the emplacement hole (profiles 1 and 4), indicate the presence of permafrost and agree with geocryological evidence [8]. The high resistivity is apparently associated with ice-rich soft sediments in the valley (peated mud with carbonate clasts, as well as sand and pebble alluvium) thermally insulated with the land cover. These features occurred outside the 160 m radius of the PUNE damage zone.

The distinct local resistivity low of 145–280 Ohm·m in the shallow subsurface near the river corresponds to a confined talik produced by the warming effect of the water. The talik was especially well detectable within the 625–715 m end of profile 4, possibly because the electrodes fell inside a small meander (having water on three sides) and were affected by shallow former riverbeds (Figure 5b).

The slope parts of profiles 1 and 4 show layered resistivity patterns of permafrost (Figure 5), which may be mainly controlled by lithology. The high-resistivity layer II (7400 to 27,000 Ohm·m), 15–20 m or locally 30 m thick, apparently corresponds to undeformed limestones at depths 10 to 25 m. Layer III, which was found to be of lower resistivity (2000–7400 Ohm·m), may represent interbedded clayey and dolomitic limestones. The frozen clay-bearing marly limestone is less resistive than the pure limestone variety, because large surface areas of clay particles increase the contents of bound (commonly saline) pore water.

The low-resistivity rocks of <2000 Ohm·m in the lower section part along profiles 1 and 4 are discussed below (Section 5.8).

### 5.2. Post-Explosion Man-Made Effects on Permafrost Temperature at the Crystal Site

Underground nuclear explosions cause direct temperature and pressure impacts on permafrost while the formation of a permeable zone leads to its further degradation as a result of fluid circulation and the ensuing heat budget changes. Additionally, the permafrost at the *Crystal* site and its surroundings was exposed to the effect of other man-made agents related to the explosion (except the first one):

- The winter road along the Ulakhan-Bysyttakh River lying on a natural surface, without subgrade; used only in winter;
- The embankment road leading to the artificial cover above the dome-shaped mound;
- The removal of land cover within the fenced territory of the *Crystal* site in December 2006;
- The fence with metal stakes put up in 2006;
- The artificial cover above the dome-shaped mound, built in 1992 and topped up in 2006.

The roads and the fence, together with the absence of land cover, led to the degradation of permafrost, whereas the artificial cover was installed with a view towards its stabilization



and even thickening to prevent the migration of radionuclides [1,4]. The contributions of these agents to the total impact on permafrost are estimated below with reference to the ERT data.

### 5.3. Winter Road

Profile 1 crosses the winter road in the river valley, whereas profile 4 crosses the winter road on a forest slope (Figure 3). In the field, we observed thermokarst-like thaw features on the winter road, which were more prominent in the river valley segment (525–560 m, profile 1), with water-filled ruts, than on the slope (280–380 m, profile 4) where the ruts were almost dry. This difference showed up in the ERT data (see insets in Figure 5): the active layer beneath the winter road was almost four times less resistive and twice as thick as that in the valley (profile 1): 74–145 Ohm·m as compared to 275–540 Ohm·m and up to 2 m as compared to 1 m, respectively. Note that the thickness of the active layer beneath winter road along profile 4 remained invariable though its resistivity was markedly lower than in the neighboring areas (Figure 5).

Thus, the thawed ground associated with the winter road was more prominent in the river valley than on the slopes grown with trees, which insulated and stabilized the permafrost. Note that the ERT method was successful in imaging this subtle difference in thermokarst features observed in the field.

### 5.4. Embankment Road

The gravel embankment road leading to the artificial cover above the dome-shaped mound caused a stronger warming effect on permafrost than the winter road (profile 4). The resistivity beneath the embankment road did not exceed 2000 Ohm·m and corresponded to an unfrozen zone that merged with another such zone in the central part of the profile (Figure 5b).

### 5.5. Removal of Land Cover

The removal of land cover, especially moss and lichen, changes the solar radiation budget and disturbs the natural thermal insulation, thus accelerating the degradation of permafrost [26]. The active layer in wildfire areas is known [26] to be twice as thick as that of undisturbed forests. In 2006, the PUNE site was cleared of soil, trees, shrubs, moss and lichen, and became spontaneously overgrown with grass, shrubs and few young larches afterwards. This led to the formation of patterned ground and thaw sinkholes, of up to 1 m depth, witnessed in 2012 and 2019, respectively, inside the fence (Figure 7).

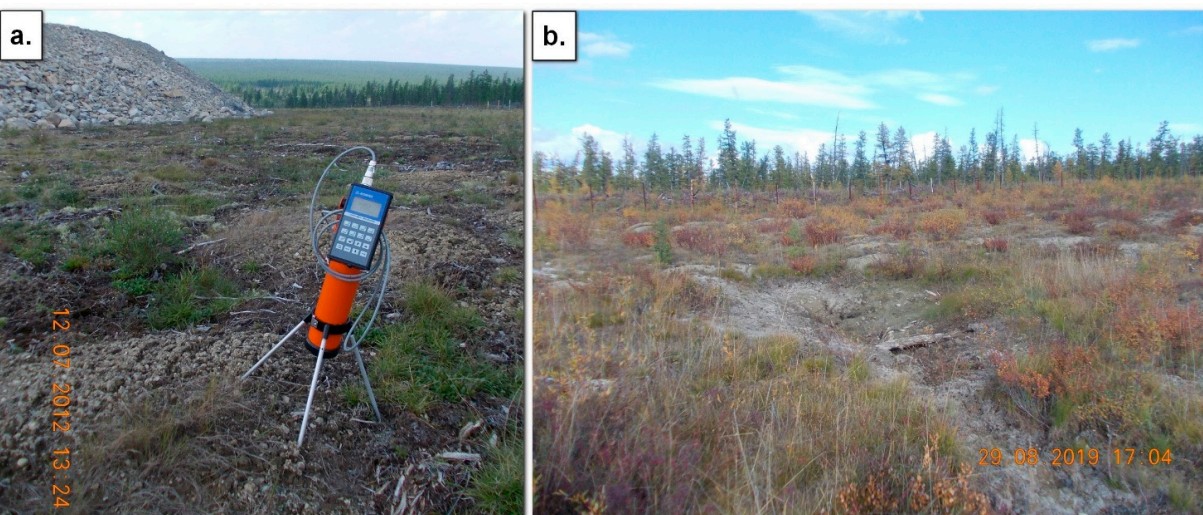

**Figure 7.** Patterned ground in the northeastern part of the *Crystal* site, in July 2012 (**a**) and in August 2019 (**b**). The pile of stones in the top left corner of the left panel is the artificial cover above the dome-shaped mound. Photograph by S. Artamonova.

The ERT data showed a resistivity of 75–540 Ohm·m in the upper 4 m within the *Crystal* site as compared to 280–1000 Ohm·m in the undisturbed slope and river valley areas and, thus, confirm that the active layer's thickness increased to 3–4 m inside the fence (Figure 6a). For example, layer II was also less resistive within than around the site: 1000–3800 Ohm·m as compared to 7400–27,000 (locally up to 100,000) Ohm·m and 3800–53,000 Ohm·m along profiles 1 and 4, respectively. Therefore, permafrost at the site is being degraded under the warming post-explosion effects. However, the fact that a line tracing distinct change in resistivity and thickness of both active layer (layer I) and layer II followed along fence, indicated that land removal in 2006 played the decisive role in the degradation of the near-surface permafrost (Figure 6a).

### 5.6. Fence

The well-pronounced features of extremely low resistivity can be related to the metal fence and the warming effect of the stakes (Figure 6). Cyclic freezing and thawing of the active layer at the site caused fence tilting, while the stakes were pushed up ~70–80 cm from their initial position in the ground (Figure 8). The ground around the stakes thawed and the sinkholes became filled with water. Therefore, the stakes conducted heat to deeper permafrost and produced thermokarst effects.

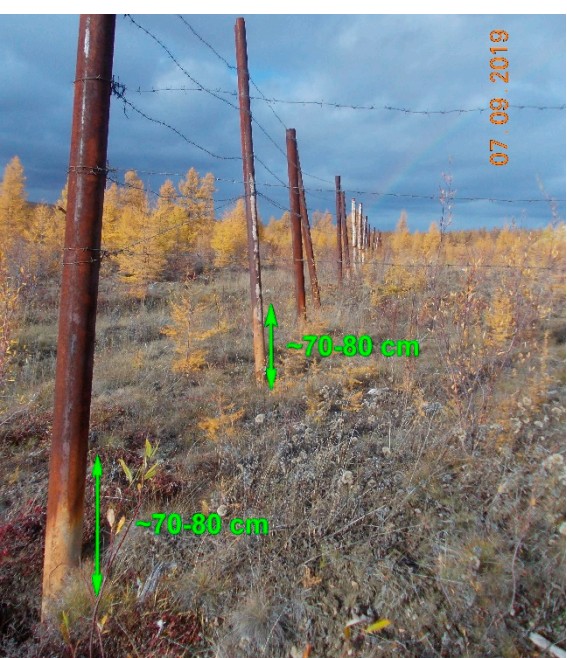

**Figure 8.** Stakes emerged from the subsided ground (September 2019).

The anomalously low resistivity values due to the fence effect were filtered from the ERT data manually. The filtering reduced the noise but did not eliminate the extreme lows beneath the fence (Figure 6); they retained the same size in the dipole–dipole data that were sensitive to vertical interfaces (Figure 6b,c).

The fence effect on the inversion results was checked by *ZondRes2D* and *Res2DInv* simulation using a layered-earth model corresponding to the natural taiga landscape (profile 1, Figure 5a). The model (Figure 9) consisted of a 2 m thick active layer on the top, with a resistivity of 100 Ohm·m; a 20 m thick resistive layer II, of 10,000 Ohm·m; and a lower-resistivity (2000 Ohm·m) layer III. The fence was simulated by two 0.1 Ohm·m conductors, of 4 m width and 2 m height, embedded into the active layer (Figure 9).

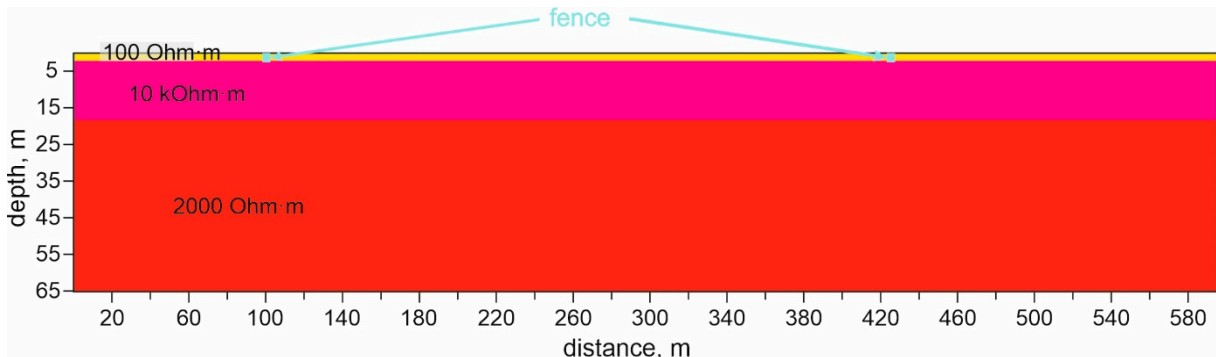

**Figure 9.** Layered-earth resistivity model with conductors (simulating a fence) embedded into the top layer.

The forward solution was obtained in *ZondRes2D* for a three-electrode array with the parameters as in the field layout (spacing 5 m, cable length 595 m), and the synthetic responses were then inverted in *Res2DInv* (Figure 10). The resulting 2D inverted resistivity section included low-resistivity zones associated with the fence, which resembled the features we observed in the field data from the *Crystal* site (Figures 6 and 10).

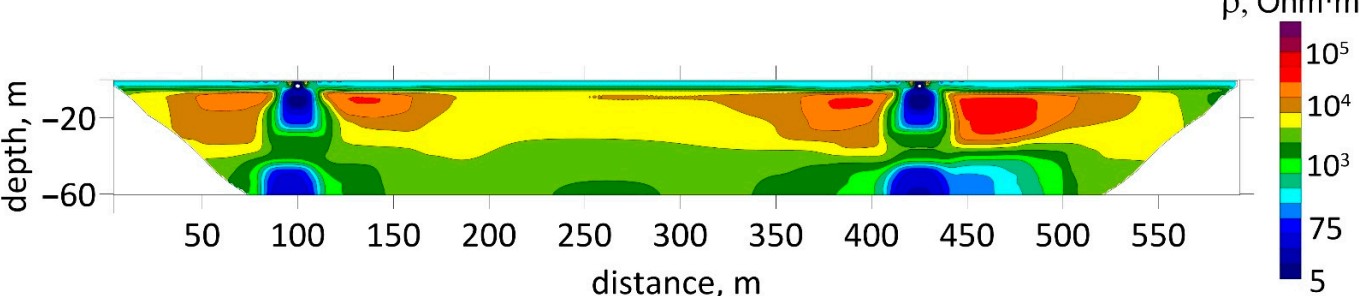

**Figure 10.** Resistivity model obtained by inversion of synthetic data, with well-defined low-resistivity features beneath the fence.

Thus, the simulation confirmed that the extremely low-resistivity features in the ERT data were mainly produced by the metal stakes and apparently included a contribution from thermokarst sinkholes around them.

*5.7. Artificial Cover*

Recall that the artificial cover, 260 m in diameter and 20 m high, was put up above the dome-shaped mound in 1992 and topped up in 2006. It was filled with dead debris (limestone, dolomite and their clayey varieties) from the Udachnaya kimberlite quarry. The cover was supposed to prevent erosion and the ensuing migration of radionuclides from the contaminated soil around the emplacement hole. Low resistivity (20 to 280 Ohm·m, locally up to 540 Ohm·m or higher) up to the cover base at 20 m in the ERT images (Figure 6b) indicates that the cover is currently unfrozen.

The heterogeneous inverted resistivity section of the underlying ground (Figure 6) may be due to different responses of lithologically diverse rocks around the emplacement hole to the explosion and its aftereffects. The resistivity behavior beneath the artificial cover requires special consideration in a separate paper, with reference to a larger amount of geophysical data, including land-based magnetic surveys.

*5.8. Warming Effect from Crystal Explosion*

Recall that the shallow PUNE *Crystal* explosion produced a damage zone consisting of a rubble chimney and crushed, cracked and spalled zones around the emplacement hole (Figure 2b). The damage zone had a radius of about 160 m on the land surface according to

estimations conducted with reference to patterns of UNE-induced rock failure studied in field experiments [1]. Thus, the damage zone occupied almost the whole site of the PUNE *Crystal*. Being permeable to heat and fluids, it can interfere with the original heat budget and, thus, might be expected to have caused permafrost degradation that was detectable in the inverted resistivity sections.

Indeed, the ERT data image low-resistivity zones (under 2000 Ohm·m) in the central segments of profiles 1 and 4 corresponded to forested slopes: 130–180 m, 230–300 m and 370–410 m in profile 1 and 80–160 m and 260–410 m in profile 4, at depths of 30–40 m and deeper (layer III). The resistivity within these zones decreased at depths from 2000 to 280 Ohm·m and was associated with unfrozen rocks, similar to the active layer's resistivity. Another thaw zone with the resistivity decreasing downward from 2000 to 75 Ohm·m appeared in the 40–70 m depth interval (Figure 5a) within the river valley (470–510 m, profile 1), 310 m north of the emplacement hole, i.e., outside the 160 m radius of the permeable damage zone. The resistivity was relatively low (3800 Ohm·m) in the river valley segment of profile 4.

Unfrozen rocks with extremely low and low resistivity (5 to 145 (locally 280) Ohm·m) were detectable below 40 m along profile 2, within both the slope and river valley segments. As in the case of profiles 1 and 4, they were not confined to the fenced area and became less resistive with depth. Therefore, the removal of the land cover alone could hardly be responsible for the local permafrost degradation.

The presence of unfrozen zones traceable at depths of 40–80 m within 200–300 m distance away from the emplacement hole, and the depthward resistivity decrease within these unfrozen zones, indicate that the rocks were exposed to the influence of a deep-seated heat and/or moisture source.

The explosion heat release was momentary and never occurred afterwards [2], and the blast source, up to now, has caused no warming effect on the ambient rocks, as no heat release from the source has occurred. In our view, the warming effect of the permeable damage zone resulting from the *Crystal* explosion changed the heat circulation patterns and produced these low-resistivity unfrozen zones. Thus, the permafrost below 40 m was subject to local warming and degradation within the 300 m distance to the emplacement hole.

The upper 30–40 m of permafrost and the active layer were also exposed to the warming effect from the damage zone; however, the warming effect from the land cover removal was more pronounced within the fenced area of the PUNE *Crystal* site. The taliks beneath the winter road and the river were deeper within the damage zone (profile 3) than in the undisturbed areas outside it. Namely, the 15–17 m deep talik within the 515–535 m segment of profile 3, corresponding to the winter road (40–540 Ohm·m zone), merged with a 10 m deep talik associated with the river (20–280 Ohm·m, segment 535–575 m).

Note that profiles 2 and 4 were located at the same distance of 200 m north and south of the emplacement hole, respectively, but their inverted resistivity sections differed markedly at all depths. The shallow resistivity patterns to a depth of 30–40 m represent different permafrost conditions in the disturbed and undisturbed landscapes (Figures 5b and 6a). Unlike the near-surface, large differences in the lower part of the sections (deeper than 40 m) were apparently due to the asymmetry of the damage zone. On the river valley segment of the profile 2, conductive, unfrozen rocks at depths of more than 40 m may be associated with the intensive damage zone north of the explosion.

## 6. Conclusions

The ERT inversion results checked against synthetic forward models yielded 2D inverted resistivity sections of permafrost to a depth of 80 m in the *Crystal* PUNE site and its vicinities. The permafrost in the natural forested slope landscape was found to have a layered resistivity structure controlled by the percentages of clay minerals in limestone. The resistivity pattern of the continuous permafrost beneath the active layer (7400 Ohm·m and higher), shown in the river valley images, with locally confined taliks under the Ulakhan-Bysyttakh River, agrees with published geocryological evidence for small rivers of the

area [8]. The active layer within the undisturbed landscapes shows up as a 1–2 m thick zone of low resistivity (tens of Ohm·m to 1000 or locally 2000 Ohm·m).

The subsurface contained local zones of unfrozen rocks at depths between 40 and 80 m in the slope and river valley profile segments, at distances of 200–300 m to the emplacement hole, whereas permafrost persisted 300–400 m away in the river valley. Therefore, the permafrost was subject to local degradation within the permeable damage zone produced by the explosion. The closer to the emplacement hole, the stronger the permafrost degradation imaged by decreasing resistivity.

The local permafrost degradation at the *Crystal* PUNE site was further affected jointly by other man-made agents, such as

- The subgrade-free winter road along the Ulakhan-Bysyttakh River;
- The gravel embankment road leading to the artificial cover;
- The land cover removal;
- The fencing.

All these effects appeared in the ERT data. The high resolution of the technique and its applicability as a useful monitoring tool at PUNE sites was confirmed, specifically by the fact that the ERT images highlighted subtle thermokarst variations visible on the winter road surface.

The relation of extreme resistivity lows detectable to a depth of 20 m beneath the fence with the metal stakes was proven by numerical simulation. The stakes conducted heat into the underlying rocks and became surrounded by sinkholes and water-logged zones of the active layer.

The removal of land cover at the *Crystal* PUNE site in December 2006 especially affected the upper 30–40 m of permafrost and the active layer. In our view, it was responsible for the almost seven-fold difference between disturbed and undisturbed areas in the resistivity of layer II to 30 m depth. It led to the formation of patterned ground, with 1 m deep sinkholes on the land surface, as well as to the thickening and reduction in resistivity of the active layer relative to that in the undisturbed areas (twice, to 4–5 m, and two to four times, to 75–540 Ohm·m, respectively).

The ETR data showed that the artificial cover composed of debris above the dome-shaped mound is currently unfrozen.

Thus, the permafrost at the *Crystal* PUNE site was found to be exposed to warming effects caused by the explosion and other related factors, which jointly caused its local degradation that is detectable in the heterogeneous inverted resistivity sections.

In the former USSR, 51 peaceful underground nuclear explosions were conducted in permafrost in taiga and tundra landscapes of northeastern Russia. It is logical to assume that at these PUNE sites, permafrost degradation and other consequences of explosions were similar to those at the *Crystal* site. Thus, the experience of research at the Crystal site will be useful for future near-surface studies at other PUNE sites.

**Author Contributions:** Conceptualization, S.A.; methodology, S.A.; software, A.S.; validation, S.A., A.S., V.P., N.K. and V.U.; fieldwork, S.A., A.S. and V.P.; writing—original draft preparation, S.A.; writing—review and editing, S.A., A.S., V.P. and N.K.; visualization, S.A. and A.S.; project administration, S.A. and V.U. All authors have read and agreed to the published version of the manuscript.

**Funding:** The study was supported by grant 18-45-140020 from the Russian Foundation for Basic Research. It was carried out on a government assignment to V.S. Sobolev Institute of Geology and Mineralogy SB RAS and on government contract F.2019.473808 as part of Program Environment Safety, Sustainable Nature Management and Forestry Development in Sakha Republic (Yakutia), for 2018–2022.

**Institutional Review Board Statement:** Ethical review and approval are not needed for our studies which did not involve humans or animals.

**Informed Consent Statement:** Not applicable.

**Data Availability Statement:** The data that support the findings of this study are available from the corresponding author, upon reasonable request.

**Acknowledgments:** The authors thank the PJSC ALROSA for administrative and technical support during field work. We would like to express our personal gratitude to Konstantin Garanin, the Chief geologist of PJSC ALROSA, and Sergey Pavlov, the chief hydrogeologist of the Udachny Division of ALROSA. The authors express thanks to Nikolai Tychkov, Senior Researcher at the IGM SB RAS, for the equipment provided. The authors thank the anonymous reviewers for their helpful comments and suggestions, which greatly improved the paper.

**Conflicts of Interest:** The authors declare no conflict of interest. The funders had no role in the design of the study; in the collection, analyses, or interpretation of data; in the writing of the manuscript, or in the decision to publish the results.

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
