# Peer review of "Shallow Permafrost at the Crystal Site of Peaceful Underground Nuclear Explosion (Yakutia, Russia): Evidence from Electrical Resistivity Tomography"

_energies, doi:10.3390/en15010301_

Round 1

Reviewer 1 Report

This article deals with an important problem in relation to the impact of underground nuclear explosions on the earth's crust. Nuclear explosions cause changes in the surrounding rocks: the rocks closest to the source are melted and vaporized, forming a containment cavity with a hard layer of molten rock.
The data and interpretations of the results are clear and well processed by the authors. . Understanding the impact of acid rain on soils and the environment is a very significant topic and deserves to be by Energies journal. This study is very well documented. This is a very good contribution for the scientific community. I therefore recommend this article for publication.

Author Response

According to recommendation of reviewer the manuscript was spell checked with the utmost care.

Reviewer 2 Report

It is very interesting case study of an underground nuclear explosion (UNE) and the data given can help us to understand what happen behind the UNE.

In my opinion. Lithology changes and the artificial changes produced in geomorphology and permafrost are well described and give us the opportunity to know and think about the changes made in other similar situations. In fact, as authors indicate, not so many publications give details about the situation and geo-environmental implications of an UNE.

It seems that this technology is useful to study the permafrost and gives data enough accurate. However, the use of electrical resistivity tomography for the study of permafrost is not new (i.e. DOI: 10.1029/2006JF000546; DOI: 10.1190/geo2015-0149.1; DOI: 10.1002/ppp.652) but the application in a place with an UNE is quite novelty.

Methods can be more detailed, unless the basement of the technique.

The results may be complemented with chemical data from radionuclides or other chemical markers, including radioactivity. But, only if available, if not, the article itself content enough information and results.

Conclusions are so related only to the case study. In fact, this could be understood. However, some general conclusions that can be applied or taking into consideration for other cases would be of great interest.

I would like to check in the text (although surely during editorial process all of this will be checked):

Line 42: “and 1988)”, the bracket.

Line 65, please check the “empty” end.

Line 147: check “with with limestone”

Line 181-182, please if possible, try to ensure that the hyphen separating a word make easy the reading of this word. In this case “man-ual” maybe ma-nual. This appears in lines: 223-224, 270-271, 433-434, 463-464. Of course I understand that it is a question about the template and the use of hyphens.

Author Response

Lines 187-188. A sentence was added about electrical resistivity tomography (ERT). This technique is not new in permafrost studies, but we were the first to apply ERT in the studies of near-surface permafrost changes initiated by an UNE.  

Lines 188, 630-634. We have cited two articles [21], [22] on the ETR application in permafrost studies.

Lines 180, 183, 619-628. We studied the radionuclides migration at the Crystal site in 2008-2009, 2012, 2018-2019; these studies were discussed in three previous articles. These articles have been added to the reference list of the manuscript [17-19].

Lines 548-552. We added a general conclusion about ERT potentialities in permafrost-related studies at UNE sites in northern regions.

Line 46 (earlier 42): “and 1988)”, the bracket was deleted

Line 69 (earlier 65): the “empty” was deleted.

Line 171 (earlier 147): second “with” deleted

Line 466: An empty space was deleted (between words “Ohm·m and “appears”)

The hyphenation of words in the manuscript is determined by the text format adopted by the Editorial board of the Journal, and we cannot influence and put extra spaces between words.

The text was checked accordingly to the comments 205-206 (earlier 179-180), 254-255 (earlier 223-224), 301-302 (earlier 268-269), 467-468 (earlier 432-433), 496-497 (earlier 461-462). 

Reviewer 3 Report

Dear Authors,

  your paper is very interesting. However, I would suggest the following two minor revisions:

  1. Please, highlight better the aim of this paper in both Abstract and Conclusions. Your results are well presented but it is not clear the purpose of this geophysical investigation and the relevance in a broader context;
  2. Please, ask an English mother-tongue to review it.

Author Response

Lines 16-19, 83-84:  We highlighted the purpose of our research in Abstract and in Introduction of the manuscript

In Novosibirsk (Russia) it is very difficult to find a reviewer who is a native speaker of English. However, the manuscript was spell checked by professional translator with the utmost care.

Reviewer 4 Report

The presented work focuses on an interesting analysis on permafrost at the Crystal PUNE. The research is well-conducted: in my opinion methodology and supporting results are well-shown.  

For me the manuscript could be fine at this stage.

thank you

Author Response

According to the recommendation the manuscript was spell checked with the utmost care.